# Development of intravenously administered synthetic RNA virus immunotherapy for the treatment of cancer

Edward M. Kennedy [1] [1] ✉, Agnieszka Denslow[1], Jacqueline Hewett[1], Lingxin Kong[1], Ana De Almeida[1], Jeffrey D. Bryant[1], Jennifer S. Lee[1], Judy Jacques[1], Sonia Feau[1], Melissa Hayes[1], Elizabeth L. McMichael[1], Daniel Wambua[1], Terry Farkaly[1], Amal A Rahmeh[1], Lauren Herschelman[1], Danielle Douglas[1], Jacob Spinale[1], Sanmit Adhikari[1], Jessica Deterling[1], Matt Scott[1], Brian B. Haines[1], Mitchell H. Finer[1], Ted T Ashburn[1], Christophe Quéva[1] & Lorena Lerner[1]

The therapeutic effectiveness of oncolytic viruses (OVs) delivered intravenously is limited by the development of neutralizing antibody responses against the virus. To circumvent this limitation and to enable repeated systemic administration of OVs, here we develop Synthetic RNA viruses consisting of a viral RNA genome (vRNA) formulated within lipid nanoparticles. For two Synthetic RNA virus drug candidates, Seneca Valley virus (SVV) and Coxsackievirus A21, we demonstrate vRNA delivery and replication, virus assembly, spread and lysis of tumor cells leading to potent anti-tumor efficacy, even in the presence of OV neutralizing antibodies in the bloodstream. Synthetic-SVV replication in tumors promotes immune cell infiltration, remodeling of the tumor microenvironment, and enhances the activity of anti-PD-1 checkpoint inhibitor. In mouse and non-human primates, Synthetic-SVV is well tolerated reaching exposure well above the requirement for anti-tumor activity. Altogether, the Synthetic RNA virus platform provides an approach that enables repeat intravenous administration of viral immunotherapy.

Oncolytic viruses (OVs) are an attractive cancer therapeutic modality that selectively kills tumor cells and inflame the tumor microenvironment (TME). Combining OVs with cancer immunotherapies has the potential to promote remodeling of the TME and activation of immune cells, enhancing the benefit of immune checkpoint inhibitors (ICIs) in poorly- or non-responsive tumors. Thus far, the therapeutic benefit of oncolytic virotherapy has been limited to intratumoral administration requiring a systemic antitumor immune response to be effective against non-injected lesions. Talimogene laherparepvec (Imlygic®)[1] has demonstrated durable responses in melanoma patients when administered intratumorally, as has Coxsackievirus A21 (CAVATAK®)[2]. Intravenous (IV) delivery of OVs may enhance efficacy by exposing all tumor sites, including small metastatic lesions, to OVs. However, the rapid development of neutralizing antibodies against the virus after IV administration likely limits exposure and infection of tumor cells after repeated dosing[3,4].

To maximize viral immunotherapy's potential, strategies to avoid neutralization must be developed. Retargeting[5,6], cell carriers[7,8], coating with polymers[9–11], and liposomes[12,13] have been utilized to shield OVs from neutralizing antibodies, but none have progressed to the clinic. Advances in nanotechnology and their use to deliver nucleic acids are paving the way for new carrier systems to overcome the challenges of IV administration of OVs[14,15].

Here, we describe the development of a nanoparticle-based delivery platform that enables repeat IV administration of viral immunotherapies. Plasmid templates are engineered and optimized for in vitro transcription (IVT) of RNA virus genomes (vRNA) that, upon formulation in lipid nanoparticles (LNP), render particles with the

[1]Oncorus, Inc., Cambridge, MA, USA. ✉e-mail: matt.kennedy@oncorus.com

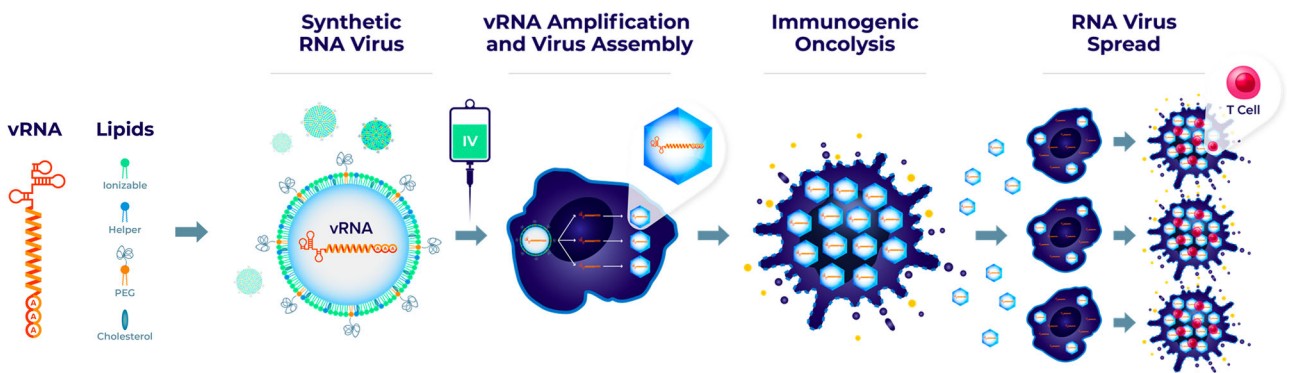

**Fig. 1 | Schematic illustration of the mechanism of action of Synthetic RNA virus.** Synthetic RNA virus is comprised of IVT-produced vRNA encapsulated within an LNP. Inside tumor cells, vRNA recapitulates all phases of the viral life cycle such that it replicates and generates a burst of infectious virions that spread locally, infecting, and killing tumor cells, thereby recruiting immune cells to the TME.

desired biophysical properties to support repeat IV administration. As all components are synthetic; we term this modality Synthetic RNA virus.

For this study, we select two picornaviruses, Seneca Valley virus (SVV) and Coxsackievirus A21 (CVA21), with well-documented oncolytic activity and clinical safety[2,16–19]. Their genomes are positive-sense single-stranded RNA and are sufficient to initiate the viral lifecycle after being introduced into a permissive tumor cell. This study reports the vRNA delivery and replication of Synthetic-SVV and Synthetic-CVA21. We show that Synthetic viruses are well tolerated and demonstrate tumor-selective viral production and spread in multiple tumor models, resulting in oncolysis and anti-tumor efficacy. We anticipate that this therapeutic platform will address the limitations associated with repeated IV administration and enhance the therapeutic potential of OVs.

## Results

### vRNA encapsulated in LNP recapitulates OV therapy when administered intravenously

To bypass neutralizing antibodies that inhibit the activity of IV-administered OVs, we developed Synthetic RNA viruses for systemic delivery of vRNA to tumor cells. Once the vRNA/LNP is internalized and released in the cytoplasm of a tumor cell, vRNA replicates and generates a burst of infectious virions that spread locally, killing adjacent tumor cells, and this promotes the recruitment of immune cells to the TME (Fig. 1). Picornaviral SVV and CVA21 vRNA constructs were validated in vitro and in vivo for their ability to produce viruses (Supplementary Figs. 1 and 2). We then optimized an LNP formulation for IV delivery with the desired biophysical properties: small size (85 nm), monodisperse, and high encapsulation efficiency (Supplementary Fig. 3a).

To ensure vRNA replication is initiated after polyprotein translation, the termini of the IVT vRNA must recapitulate those of the RNA viral genome. These termini are generated during IVT by an optimized 5′ ribozyme and runoff template produced by cleavage by a Type IIS restriction enzyme at the 3′ termini. To enhance the potency of Synthetic-SVV, we introduced modifications in the internal ribosome entry site (IRES)[20] and a mutation in VP2 that improves viral entry[21]. These modifications enhanced its efficacy compared with the SVV-001 vRNA[22] (Supplementary Fig. 3b, c). At the end of the study, SVV replication was detected in tumor cells and not in liver tissue (Supplementary Fig. 3d), indicating that systemic distribution of vRNA/LNP led to viral replication in permissive tumor cells and not in healthy tissues.

### Intravenous administration of Synthetic-SVV inhibits tumor growth in an SCLC cancer model

Synthetic-SVV administration resulted in significant tumor growth inhibition (TGI) of NCI-H446 small cell lung cancer (SCLC) xenografts (Fig. 2a), including regression of very large tumors (>500 mm³, Supplementary Fig. 4a) without significant body weight loss (Fig. 2b). Administration of Synthetic-SVV-Neg (a replication-incompetent vRNA) did not inhibit tumor growth (Fig. 2a), demonstrating that TGI is associated with SVV replication. Three days after IV injection of Synthetic-SVV or SVV virions as a positive control, robust fluorescent in situ hybridization (FISH) signal in tumor was detected from both the positive and negative-sense RNA strands (Supplementary Fig. 4b). Detection of negative-strand RNA, a template for picornavirus positive-strand RNA genome, unequivocally confirms viral RNA replication. In tumors of mice dosed with non-replicating Synthetic-SVV-Neg, only low levels of positive-strand RNA was detected, likely associated with residual vRNA/LNP in blood vessels (Supplementary Fig. 4b).

### IV administration of Synthetic-SVV leads to rapid kinetics of intratumoral viral replication

To explore the dose-response relationship after IV administration of Synthetic-SVV, we performed a dose titration. Potent TGI of NCI-H466 xenograft tumors was observed at all dose levels, with maximal TGI achieved with 0.1 mg/kg or above (Supplementary Fig. 4c). A 0.1mg/kg dose was thus selected to characterize viral replication kinetics. After a single IV administration of Synthetic-SVV, tumors were harvested at multiple time points and analyzed by RT-qPCR and FISH. Viral negative-strand RNA and intratumoral virions were detected as early as 3 days and reached a plateau in most tumors at 7 days post treatment. Remarkably, sustained SVV replication was detected up to 21 days after administering a single low dose of Synthetic-SVV (Fig. 2c). These findings were largely recapitulated with FISH detection of SVV positive- and negative-RNA strands, with the FISH signal widely distributed in the tumor and peaking by Day 10 (Fig. 2d).

### The Synthetic vRNA platform is applicable to other picornaviruses

CVA21 was selected as a second candidate Synthetic RNA virus, based on its oncolytic properties, favorable IV tolerability in cancer patients[2,23,24], and distinct tumor tropism compared to SVV[16,24]. CVA21 IVT transcribed vRNA formulated with the same lipid composition as utilized for Synthetic-SVV yielded nanoparticles with similarly desirable biophysical properties. Complete tumor regression at low dose levels was observed in the SK-MEL-28 melanoma model (Fig. 2e), without significant body weight loss (Fig. 2f).

**Tolerability of Synthetic RNA viruses in permissive immuno-competent mice.** Tolerability of Synthetic-SVV virus was evaluated in immunocompetent A/J mice permissive to SVV infection[22]. Intravenous administration of Synthetic-SVV at 3 mg/kg was well tolerated when

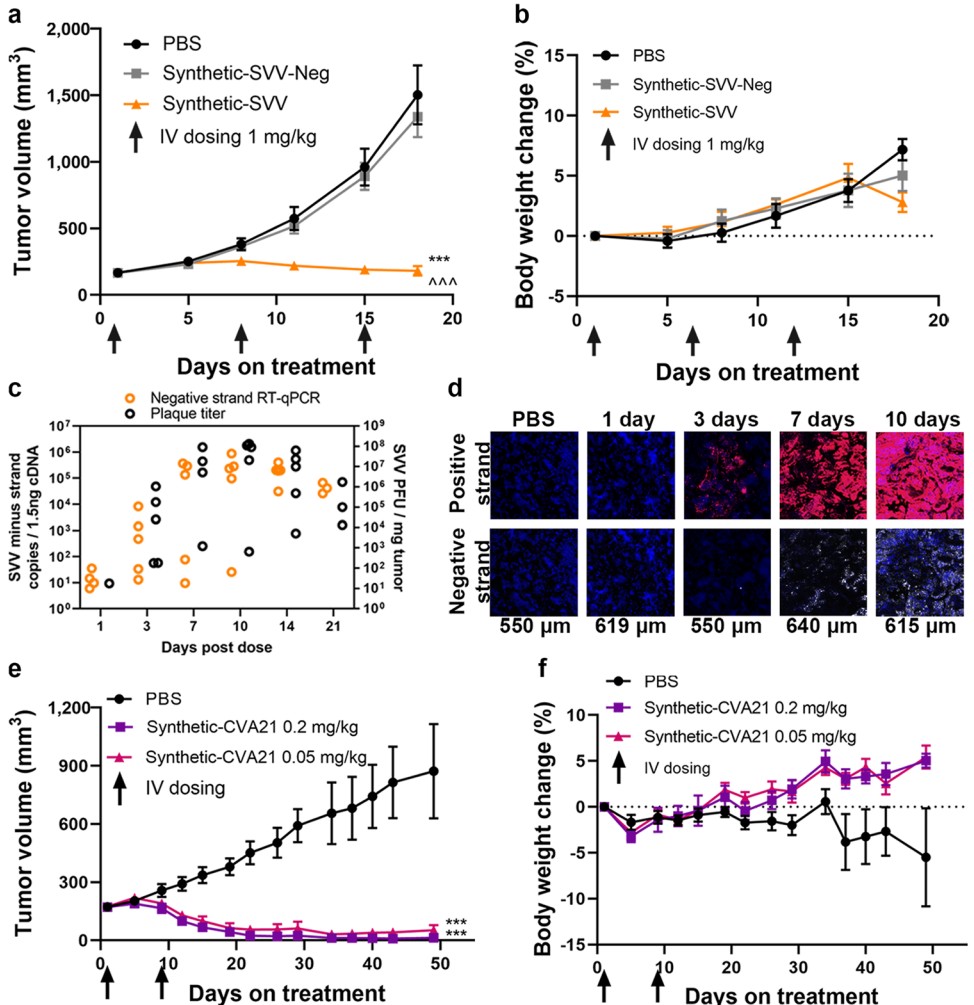

**Fig. 2 | Intravenous Synthetic RNA virus administration demonstrates viral replication in tumors and mediates tumor regression. a–d** Athymic nude mice implanted subcutaneously with NCI-H446 SCLC xenograft tumors. **a, b** Mice were treated via IV administration with either vehicle control (PBS), Synthetic-SVV-Neg, or Synthetic-SVV on Days 1, 8, and 15, at 1.0 mg/kg ($n = 8$ per group). **a** Tumor volume (mm³) and **b** body weight changes (%) were monitored. Tumor growth (mm³) and body weight changes (%) were monitored. Data are reported as mean ± s.e.m. Statistical significance was determined using a mixed linear model ***$p < 0.001$ vs. PBS and ^^^$p < 0.001$ vs. Synthetic-SVV-Neg. **c, d** Replication of Synthetic-SVV in NCI-H446 tumors after a single IV dose of 0.1 mg/kg SVV. **c** SVV negative-strand RNA levels were determined using RT-qPCR ($n = 5$ per time point). SVV infections particles (PFU) were determined by plaque assay. $n = 3$ independent tumor samples were assessed per timepoint. **d** FISH specific for SVV positive (red) and negative (white) RNA strands are shown for NCI-H466 tumor sections. Nuclei were labeled with 4′,6-diamidino-2-phenylindole (DAPI). These images are representative of four independent samples. Scale of the panel is indicated under the image. **e, f** Athymic nude mice ($n = 7$ per group) implanted subcutaneously with SK-MEL-28 human melanoma tumors were treated by IV administration with either vehicle control (PBS) or Synthetic-CVA21 on Days 1 and 8 at 2 different doses, 0.2 mg/kg or 0.05 mg/kg. **e** Tumor growth (mm³) and **f** body weight changes (%) were monitored. Data are reported as mean ± s.e.m. Statistical significance was determined using a mixed linear model ***$p < 0.001$ vs. PBS. Source data are provided as a Source Data file.

administered to A/J mice bearing a syngeneic neuroblastoma tumors, N1E-115[25]. No significant adverse body weight change, clinical signs, or histopathology findings (Supplementary Fig. 5a) were observed. Quantification of OC by liquid chromatography-mass spectrometry (LC-MS) showed broad LNP distribution to all analyzed tissues at 30 min post-dose (Supplementary Fig. 5b) followed by fast clearance as evidenced by lack of OC detection at 24 hr. There were no changes in clinical chemistry, including liver function (Supplementary Fig. 5c–e). A transient elevation of proinflammatory cytokines was observed (Supplementary Fig. 5f–j), as reported for other systemically administered LNPs[26], but it was not accompanied by complement activation (Supplementary Fig. 5k). In naïve A/J mice, Synthetic-SVV showed minimal and transient viral replication in lung, spleen, and liver (Supplementary Fig. 5l).

Synthetic-CVA21 was also well tolerated in a transgenic mouse expressing the human ICAM1 (huICAM1) gene, the cellular receptor for CVA21, and known to be sensitive to CVA21 infection[27,28]. IV administration of Synthetic-CVA21 did not lead to any adverse clinical signs (Supplementary Fig. 6a). Histopathology findings were limited to mild microscopic changes in liver, attributed primarily to the treatment with LNP formulation. Low levels of CVA21 replication were detected by RT-qPCR and by plaque titer assay in the spleen, liver, lung, heart, and kidney 2 days after dosing. However, negative-strand RNA or CVA21-virions were undetectable at 7 days (Supplementary Fig. 6b, c), indicating that the mice had cleared CVA21 infection. These data suggest that Synthetic-SVV and -CVA21 are well tolerated in mouse models known to be permissive to these viruses at dose levels above those necessary to elicit tumor regression.

**Tolerability of synthetic-SVV in cynomolgus monkeys.** The tolerability of Synthetic-SVV was further evaluated in non-human primates (NHP) dosed three times every two weeks with 1 mg/kg Synthetic-SVV.

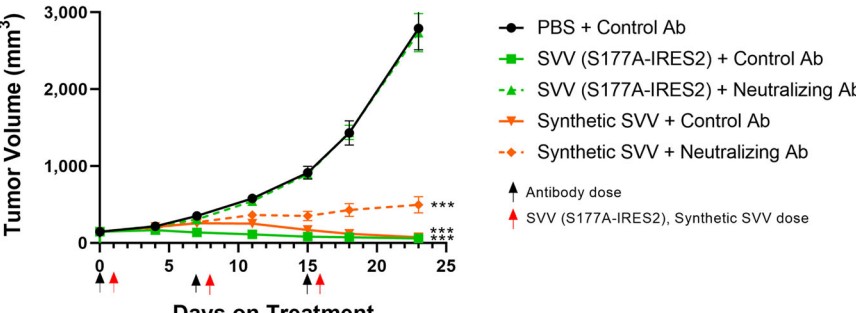

**Fig. 3 | Synthetic-SVV is efficacious in the presence of circulating SVV-neutralizing antibodies.** Anti-tumor efficacy of Synthetic SVV-derived virus SVV(S177A-IRES2) and Synthetic-SVV as assessed by tumor volume (mm³) was evaluated in NCI-H446 tumor bearing mice after injection of control or neutralizing anti-SVV rabbit serum. Mice that were passively immunized with SVV antisera received 3 IV injections of either $10^6$ PFU of SVV(S177A-IRES2) virions or 0.1 mg/kg Synthetic-SVV ($n$ = 10 per group). Data are presented as mean values +/− SEM. Statistical significance was determined using a mixed linear model. $p < 0.001$ vs PBS, SVV(S177A-IRES2) & Control Ab. Source data are provided as a Source Data file.

No clinical signs, significant effects on body weights were observed (Supplementary Fig. 7a). Pharmacokinetic analysis of the LNP lipid component indicated that the exposure ($AUC_{0-\infty}$) reached in NHP plasma was 85x fold over the exposure achieved in athymic nude mice dosed with maximally efficacious dose level in the NCI-H446 SCLC model (Supplementary Fig. 4a and Table 1). Systemic administration of Synthetic-SVV distributed the vRNA to all tissues examined, including the spleen, liver, kidney, muscle, lung, and brain, 24 h after the third dose (Supplementary Fig. 7b), mirroring the broad biodistribution for the LNP lipid component observed in tumor bearing mouse (Supplementary Fig. 5b).

Furthermore, no histological findings were observed in NHP. Only minor and transient elevation of liver chemistry parameters were observed (Supplementary Fig. 7c–e). Plasma proinflammatory cytokines, including IL-6 and MCP-1 and complement activity were transiently increased within 2–24 h post intravenous dose (Supplementary Fig. 7f–h). These results demonstrate the tolerability of repeat dose Synthetic-SVV in cynomolgus monkeys at a dose level that exceeds exposures necessary to elicit potent antitumor activity.

### In vivo activity of Synthetic RNA virus therapy is not altered by virus-neutralizing antibodies

We postulated that the delivery of vRNA/LNP would be resistant to virus neutralizing antibodies in circulation. To test this hypothesis, we passively immunized NCI-H466 bearing immune-deficient mice with either a control rabbit serum or a rabbit anti-SVV serum with confirmed neutralization potency for the SVV-001, SVV(S177A), and SVV(S177A-IRES2) viruses, the latter being encoded by the vRNA in Synthetic-SVV (Supplementary Fig. 8a–c). We then compared anti-tumor efficacy of 0.1 mg/kg Synthetic-SVV or SVV(S177A-IRES2) virions (Fig. 3). Control serum-treated animals dosed with SVV(S177A-IRES2) or Synthetic-SVV exhibited potent antitumor activity relative to vehicle control. By contrast, in mice pre-treated with anti-SVV serum, the efficacy of IV-administered SVV(S177A-IRES2) was completely abrogated, while Synthetic-SVV remained potent at inhibiting the growth of NCI-H466 xenografts. Similar abrogation of the activity of the clinically evaluated SVV-001 was observed in presence of neutralizing antibodies while Synthetic-SVV remained active (Supplementary Fig. 8d)

### Synthetic-SVV prolongs the survival of mice bearing orthotopic SCLC tumors

We then sought to evaluate Synthetic-SVV activity in models that may be more representative of human SCLC than the subcutaneous xenograft models. SCLC is an appropriate indication for the clinical development of Synthetic-SVV, given that SVV has an established tropism for tumors of neuroendocrine origin[16,22]. In mice bearing NCI-H82 tumors orthotopically implemented in the lung, two IV administrations of Synthetic-SVV triggered SVV replication (Supplementary Fig. 9a) and nearly doubled the survival vs. control arms (Fig. 4a). Immunohistochemistry (IHC) for human Delta-like ligand 3 (hDLL3), a neuroendocrine marker highly expressed in NCI-H82 cells (Supplementary Fig. 9b), was utilized to quantify tumor burden. Assessment of tissues collected 10 days after treatment revealed a significant reduction of tumor burden and extensive central tumor necrosis in the lung of mice treated with Synthetic-SVV (Fig. 4b, c).

### Synthetic-SVV is efficacious in SCLC PDX and GEMM tumor models

Patient-derived xenograft (PDX) and genetically engineered mouse models (GEMMs) are thought to better represent the heterogeneity and genetic alteration of human cancers[29]. Given that SCLC is known for its heterogeneity leading to the emergence of treatment resistant disease and recurrence[30–32], we evaluated the ability of Synthetic-SVV to replicate in an SCLC PDX model. Mice bearing an SCLC-PDX model were dosed IV with Synthetic-SVV, Synthetic-SVV-Neg, or vehicle and intratumorally with SVV-001 virions as a positive control. Similar levels of viral replication were observed when Synthetic-SVV was administered systemically, or SVV-001 virions were dosed intratumorally (Supplementary Fig. 9c). Furthermore, potent anti-tumor activity was observed after Synthetic-SVV administration (Fig. 4d).

Tumors arising from GEMMs closely mimic their human counterparts' molecular features, tumor heterogeneity, and histopathology. We evaluated the efficacy of Synthetic-SVV in a transplantable SCLC GEMM model[33] ($Rb1^{fl/fl}Trp53^{fl/fl}Myc^{LSL/LSL}$) with histopathologic and transcriptional profiles similar to human SCLC. Synthetic-SVV significantly inhibited tumor growth (Fig. 4e). Tumors collected 5 days after dosing demonstrated high level of SVV replication (Supplementary Fig. 9d). Additionally, we profiled the changes in the immune microenvironment in this model by Nanostring PanCancer IO 360™ Panel. We observed notable increased scores associated with cytotoxicity, interferon signaling, and lymphoid compartment pathways in tumors of mice treated with Synthetic-SVV vs. vehicle control (Supplementary Fig. 9e), indicating that in situ SVV replication led to more inflamed TME.

### Combination therapy with PD-1 antagonist enhanced Synthetic-SVV activity

The change in the TME elicited by Synthetic-SVV was characterized in the syngeneic N1E-115 tumor, a model poorly infiltrated by immune-cells (Supplementary Fig. 10). Administration of Synthetic-SVV led to a significant increase in the recruitment of CD8 T-cells and a trend for CD4 T-cells and NK-cells (Fig. 5a and Supplementary Fig. 11a). Regulatory T-cells (Treg) numbers did not increase in tumors, leading to an elevated CD8/Treg ratio that has been associated with improved

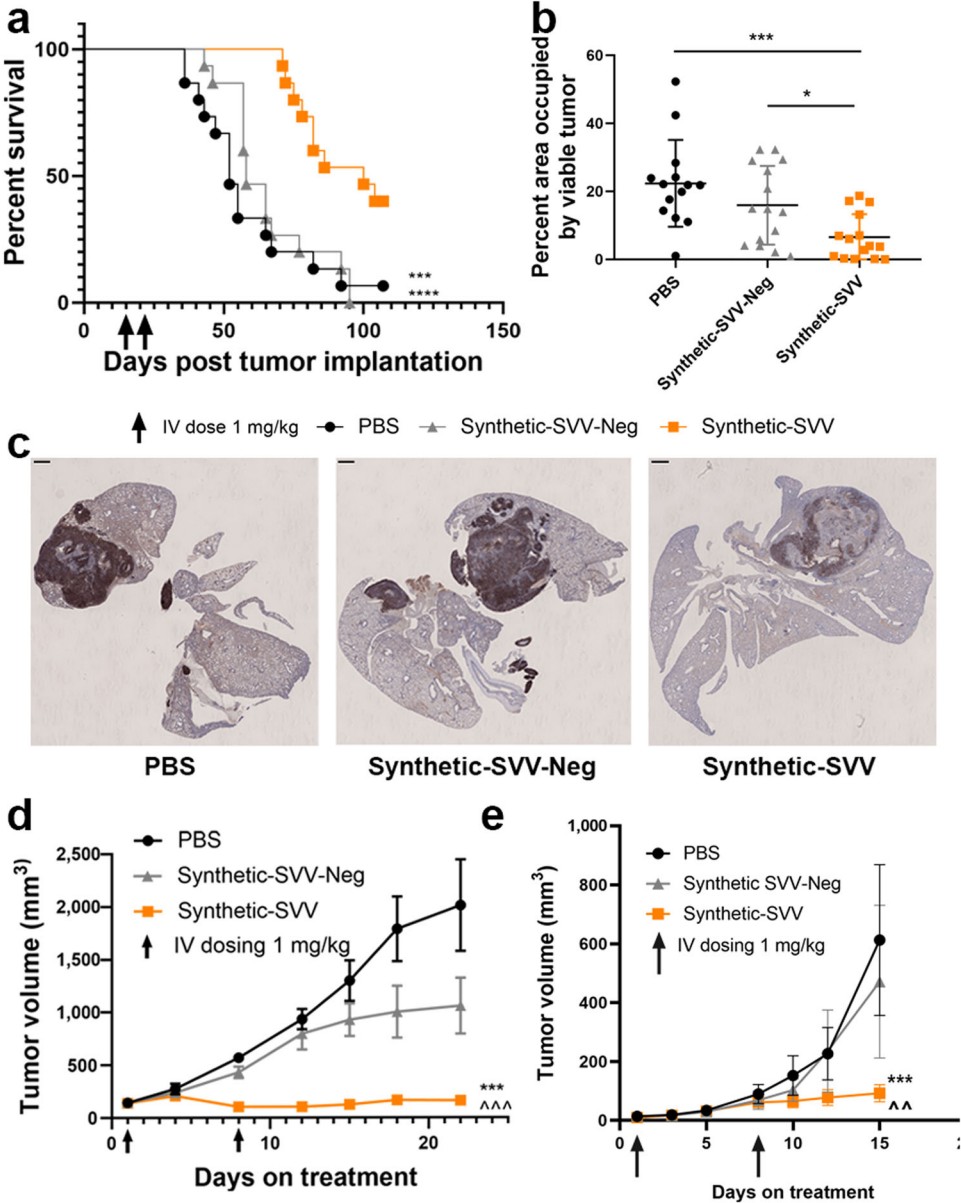

**Fig. 4 | Synthetic-SVV treatment is efficacious and prolongs survival in an orthotopic SCLC model. a–c** NCI-H82 tumors were orthotopically implanted in athymic nude mice. Animals were dosed either with PBS, 1.0 mg/kg Synthetic-SVV-Neg, or 1.0 mg/kg Synthetic-SVV on Days 15 and 22 post tumor inoculation ($n = 15$ per group). **a** Survival analysis was assessed using log-rank (Mantel–Cox) test, ***$p = 0.003$; ****$p < 0.0001$ vs. Synthetic-SVV. **b** Morphometric analysis of hDLL3 positive area of pulmonary and extrapulmonary tissue occupied by viable tumor in lungs collected from mice 10 days post-treatment. Statistical significance was determined using two-tailed paired T-test, *$p = 0.011$; ***$p = 0.0003$. **c** Representative images of hDLL3 staining as determined in 15 individual samples/ treatment arm. Scale bars in the upper left are 1000 μm. **d** NOD/SCID mice were

implanted subcutaneously with SCLC PDX tumors (Crown Bioscience, San Diego, CA) and dosed by IV administration with either vehicle control (PBS), Synthetic-SVV-Neg, or Synthetic-SVV 1 mg/kg on Days 1 and 8 ($n = 8$ per group). Tumor volume (mm³) was monitored at various time points. **e** RPM mice were implanted subcutaneously with SCLC GEMM primary culture cells and dosed by IV administration with either vehicle control (PBS), Synthetic-SVV-Neg, or Synthetic-SVV 1 mg/ kg on Days 1 and 8 ($n = 8$ per group). Tumor volume (mm³) was monitored at various time points. **d, e** Data are presented as mean values +/− SD. Statistical significance was determined using mixed linear model, ***$p < 0.001$ vs. PBS, and ^^$p < 0.01$; ^^^$p < 0.001$ vs. Synthetic-SVV-Neg. Source data are provided as a Source Data file.

clinical benefit with anti-PD-1 therapy[34] (Fig. 5b). The CD8 T-cells showed an activated phenotype, with upregulated CTLA4 and PD-1 (Fig. 5c). Both short-lived effector cells (SLEC, CD8+, CD127, KLRG1+) and memory precursor effector cells (MPEC, CD8+, CD127+, KLRG1−) were increased with Synthetic-SVV compared with control (Fig. 5d). Tumor-associated macrophages were also profiled, and an increased M1 (phagocytic, CD206-CD86+)/M2 (proinflammatory, CD206+CD86−) ratio was observed (Fig. 5e and Supplementary Fig. 11b). The number of M1 macrophages (Fig. 5f) and tumor cells expressing PD-1 ligand (PD-L1) were also significantly increased

(Fig. 5g). Concordant results were obtained when tumors were analyzed by NanoString PanCancer IO 360™ Panel (Supplemental Data 1). Synthetic-SVV treatment significantly increased pathways associated with immune cells recruitment and activation (Supplementary Fig. 12). These data indicate that Synthetic-SVV promotes immune cells recruitment and antitumor immunity. Moreover, two weekly doses of Synthetic-SVV led to significant TGI (Supplementary Fig. 13).

The increased inflammation in the TME and upregulation of PD-L1 on N1E-115 tumor cells and tumor-associated macrophages provided a strong rationale for combining Synthetic-SVV with a PD-1 antagonist.

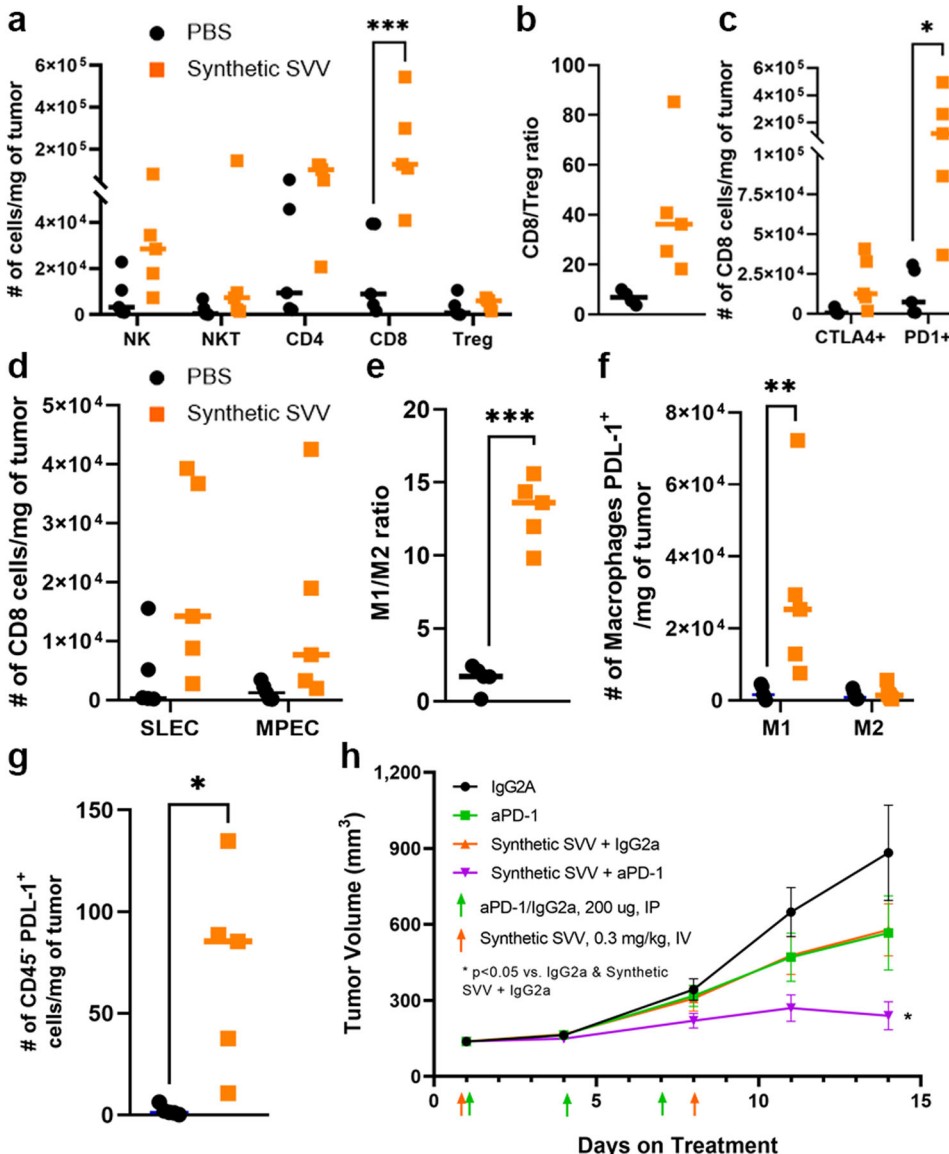

**Fig. 5 | Synthetic-SVV promotes immune cell recruitment and enhances the activity of PD-1 inhibitor.** **a**–**g** A/J mice (n = 5 per group) implanted subcutaneously with syngeneic neuroendocrine N1E-115 tumors. Mice were treated by IV administration with either PBS vehicle control or 1 mg/kg Synthetic-SVV on Days 1 and 8. Tumors were collected 6 days after the second dose. **a** Number of NK, NKT, CD4, CD8, and Treg cells per mg of tumor. For CD8 p = 0.0002 with a two-way ANOVA test **b** CD8/Treg ratio. **c** Number of CD8 T cells per mg of tumor that express CTLA-4 or PD-1. For PD1 p = 0.01 with a two-way ANOVA test. **d** Number of CD8 T cells per mg of tumor that are SLEC (CD127-KLRG1+) or MPEC (CD127+-KLRG1−). **e** Ratio of M1/M2 macrophages. p = 0.0002 with a two-tailed paired students T test. **f** Number of M1 and M2 macrophages PD-L1+ per mg of tumor. p = 0.01 with a two-tailed paired students T test. **g** Number of CD45- PD-L1+ cells per mg of tumor. p = 0.01 with a two-tailed paired students T test. **h** A/J mice (n = 10 per group) were implanted subcutaneously with N1E-115 tumors and treated by IV administration with either Synthetic-SVV on Day 1 and 8 (0.3 mg/kg), and IP administration control antibody (mouse IgG2a) or anti-PD-1 antibody on Days 1, 4, and 7 (200 μg) as indicated. Tumor volume (mm³) was monitored. Data are reported as mean ± s.e.m. Statistical significance was determined using a mixed linear model was applied, *p = 0.01vs. IgG2a, *p = 0.03 vs. Synthetic-SVV/IgG2a. **h** Source data are provided as a Source Data file.

Surprisingly for this cold tumor model, anti-PD-1 antibody yielded a modest but significant TGI, similar to a suboptimal dose of Synthetic-SVV; however, their combination provided a significant superior efficacy than each single agent (Fig. 5h).

## Discussion

This report describes the design and development of Synthetic RNA viruses for the systemic treatment of cancer. Intravenous delivery of the vRNA genomes for two picornaviruses, SVV and CVA21, formulated in LNPs, was well tolerated and elicited tumor-specific in situ production of OVs, immune cell recruitment, and ultimately tumor destruction. Efficacy was observed in multiple cancer models, including xenografts, PDX, GEMM, and syngeneic models. Moreover, a survival

benefit was observed in an orthotopic SCLC tumor model. Synthetic-SVV remained efficacious even in the presence of circulating virus-specific neutralizing antibody and was further potentiated by the combination with an inhibitor of the immune checkpoint PD-1.

Clinical stage OVs, including those with no or little prior exposure in humans such as SVV, CVA21, or enadenotucirev, a non-naturally occurring adenovirus, have reported neutralization after repeated IV dosing, likely limiting the window of effectiveness to a short time period[16,17,23,35,36]. We showed here that, in contrast to the in vivo activity of SVV virions, the activity of Synthetic-SVV remained potent even in the presence of SVV neutralizing antibody. While both treatments utilize same viral genome it is important to acknowledge that Synthetic Virus modality differs significantly from its picornaviral parent with

respect to stability, biodistribution, endosomal escape, and initial entry tropism; and yet the dose of Synthetic-SVV delivered to the tumor was still sufficient to elicit strong antitumor efficacy in the presence of neutralizing antisera. We hypothesize that once viral genome is delivered and replication is initiated in permissive tumor cells, the antibody concentration in the interstitial tumor milieu does not reach a threshold required to inhibit cell-to-cell spread during viral infection. This data is in line with the lack of impact of preimmunization on oncolytic HSV preclinical and clinical activity after intratumoral injection[1,37]. Therefore, our strategy is anticipated to overcome the central challenge in the field of OVs by enabling repeat IV administration, thus providing an opportunity for all tumor lesions within a patient to be exposed to the therapeutic agent while evading neutralization. Finally, we expect that systemic administration of Synthetic RNA viruses will benefit patients with disseminated disease and patients for which conducting safe repeat intralesional OV injections is challenging. The treatment of lung cancer is particularly interesting in this regard as the two Synthetic RNA viruses that we describe here, SVV and CVA21, have a tropism for SCLC and NSCLC, respectively.

SVV has a known tropism for SCLC[17,22], and this is confirmed by our data showing anti-tumor activity in multiple SCLC models. CVA21 tumor tropism is driven by the expression of its entry receptor ICAM1, which is highly expressed in NSCLC and other tumor indications[38,39]. In addition to its entry receptor, viral tropism is also limited post-entry by Type I IFN and viral restriction factors within the infected cell. These are essential to consider in the context of the Synthetic RNA virus delivery platform, as it uses LNP to initially bypass the viral entry receptor. While IV administration of Synthetic RNA virus led to broad tissue distribution, minimal and transient detection of SVV or CVA21 replication was observed in normal tissues from sensitive mouse strains. These results demonstrate selective tumor replication and elimination by host anti-viral response in normal tissues. Overall, Synthetic RNA viruses were well tolerated after a single or multiple IV doses in mice and non-human primates, as were the mRNA/LNP formulations encoding protein therapeutics described by others in preclinical[40] and clinical studies[41].

The mode of action of OVs involves the direct killing of cancer cells and the stimulation of anti-tumor immunity[42]. The treatment of SCLC and NSCLC might particularly benefit from repeat IV delivery of OVs as enabled by the Synthetic RNA virus platform, as safe intralesional injections are challenging in lung cancer. SCLC and NSCLC are known for a high tumor mutational burden[43,44], for their sensitivity to immune checkpoint inhibitors, and for being able to replicate SVV and CVA21, respectively. While the potent anti-tumor activity of Synthetic RNA virus in human tumor models xenografted in immunocompromised mice is likely due to the unconstrained spread of infection within the tumor and oncolysis, the infection may likely be limited by immune cells recognizing and eliminating infected cells in an immune competent host. Immune sensing of viral replication may, in turn, facilitate the efficacy of Synthetic RNA viruses by enhancing immune cells recruitment and activation. As observed for other OVs[37], Synthetic-SVV remodeled the TME and increased the number of CD8 T-cells and M1 phagocytic macrophages. Upregulation of PD-L1 on tumor and myeloid cells provided the rationale for combining Synthetic-SVV with anti-PD-1, demonstrating an improved therapeutic benefit above each monotherapy. The benefit of the PD-(L)1 antagonists currently approved for the treatment of SCLC and NSCLC are limited to a small percentage of patients; our data suggest that combining them with Synthetic RNA viruses may improve outcomes in these patients.

Synthetic RNA viruses represent a therapeutic modality for cancer treatment with the potential to transform the current intralesional injection paradigm for OVs into a convenient IV repeat administration. This modality has the potential to expose all tumor lesions within a patient to a potently oncolytic living drug that can infect and spread in tumors while stimulating anti-tumor immune responses. Synthetic-SVV and Synthetic-CVA21 have potent activity in preclinical models, even in the presence of neutralizing antibodies, and are well tolerated in rodent and non-human primates. These data support the progression of Synthetic-SVV and -CVA21 to clinical trials for the treatment of lung cancer and other permissive tumor indications as monotherapy or in combination with ICIs-containing standard of care regimen.

## Methods

All experimental procedures in animal models, including mouse and non-human primates were reviewed and approved by the respective Institutional Animal Care and Use Committees, followed NIH guidelines as specified in the Guide for the Care and Use of Laboratory Animals, 8th Edition. NHP studies were conducted in an Association for Assessment and Accreditation of Laboratory Animal Care (AAALAC)-accredited facility.

### Data collection and analysis

RT-qPCR were collected using QuantStudio 6 Pro Real-time PCR system by ThermoFisher (Applied Biosystems). Immunohistochemistry (IHC) sections were digitally scanned using the Hamamatsu Nanozoomer, Spectra Max i3X minimax imaging cytometer was utilized to quantify florescence, and lipid nanoparticle (LNP) characterization was performed with Zetasizer Nano ZS, BO LSRFortessa using BO FACSDiva v8.0.3 software. For the immunofluorescent (IF) assays, images were acquired on the 30 Histec Panoramic 250 scanner. For NanoString, samples were analyzed using nCounter digital analyzer. OC lipid concentration was determined via LC-MS analysis. A Waters Zevo TQs system equipped with a MRM detector was used for the chromatographic step.

Microsoft Excel (Office 365), Graphpad Prism v9, FlowJo vl0.7.1, and 30 Histec Caseviewer 2.4 (CaseViewer- 3DHISTECH Ltd.) were used to analyze FISH slides. QuPath was used to evaluate DLL3 IHC. Data collection was carried out on the nCounter Digital Analyzer (NanoString Technologies).

### Cell lines

Cell lines NCI-H466 (HTB-171), NCI-H82 (HTB-175), NCI-H1299 (CRL-5803), SK-MEL-28 (HTB-72), CT26 (RL-2638), 4T1 (CRL-2539), and N1E-115 (CRL-2263) were all purchased from ATCC (Gaithersburg, MD). CT-2A-luc (SCC195) and MCC14/2 (10092303) were purchased from Millipore Sigma (Burlington, MA). Murine colon adenocarcinoma cell line MC38 was kindly donated by Prof. Joseph Glorioso from the University of Pittsburgh. NCI-H446, NCI-H82, NCI-H1299, SK-MEL-28, MCC14/2, CT26, and 4T1 cell cultures were all maintained in RPMI-1640 medium (Gibco, Gaithersburg, MD) supplemented with 10% heat-inactivated FBS (Gibco, Gaithersburg, MD) and 1% penicillin/streptomycin (Gibco, Gaithersburg, MD). MC38, CT-2A-luc, and N1E-115 cells were cultured in DMEM medium (Gibco, Gaithersburg, MD) supplemented with 10% heat-inactivated FBS and 1% penicillin/streptomycin. All cell cultures were incubated in a humidified atmosphere with 5% $CO_2$ at 37 °C.

### IVT template design and construction

Picornaviral positive-strand sequences were obtained from NCBI (SVV-001[45] (GenBank: DQ641257) and CVA21[46] (Genbank: AF546702.1). Custom ribozymes were designed for the 5′ of the viral template, and 30 nucleotide poly adenosine (pA) sequences were added to the 3′ end, followed by either SapI (SVV) or BsmBI (CVA21) restriction site to generate polythymidine templates of the appropriate length after linearization. The SVV-S177A mutation was introduced by point mutagenesis for the SVV-S177A virus utilized in Supplementary Fig. 8, and the Synthetic SVV virus, SVV(S177A-IRES2) is comprised SVV-001 including this mutation and an improved IRES, SVV IRES2. This sequence is derived from SVA/Canada/MB/NCFAD-104-1/2015 (GenBank: KY486156). The IVT templates were constructed with synthetic

dsDNA fragments (IDT Geneblocks, Genscript, Piscataway, NJ) and Gibson assembly (NEBuilder HiFi DNA Assembly Master Mix, Catalog #E2621L, NEB, Ipswich, MA) following the manufacturer protocol. These constructs were sequenced end to end, linearized with either SapI (SVV) or BsmBI (CVA21) (NEB SapI # R0569L, BsmBI # R0739L, Ipswich, MA), and research-grade IVTs (NEB HiScribe T7 High Yield RNA Synthesis Kit Catalog #E2040S, Ipswich, MA) were performed to ensure viral kickoff. All viral stocks used in this work were obtained with these reverse genetics systems.

## IVT and LNP formulation
Large-scale IVTs (20-100 mg) were performed at Aldevron (Fargo, ND) and purified by diafiltration. In some instances, IVTs (1–50 mL) were performed internally using a heating block (Eppendorf Thermomixer, Hamburg, Germany) at 37 °C for 2.5 h using a buffer containing magnesium acetate, Tris-HCl (EMD-Millipore, Danvers, MA), TCEP (EMD Millipore Danvers, MA), equimolar NTPs (Thermo Fisher Scientific, Waltham, MA), inorganic pyrophosphatase (NEB Ipswich, MA), and T7 RNAP (NEB, Ipswich, MA). DNase I (NEB, Ipswich, MA) was added at the end of the IVT for 30 min and quenched with EDTA (Thermo Fisher Scientific, Waltham, MA). Next, tangential flow filtration (TFF) was performed using a 100 kDa mPES membrane (Repligen, Marlborough, MA) and diafiltered using water. The TFF retentate was subsequently salt adjusted and loaded onto an Oligo-dT chromatography column (BIA Separations, Ajdovščina, Slovenia). RNA containing a pA tail was eluted using water. A final TFF step was performed as a desalting step to ensure the desired RNA concentration (1.0 mg/mL) and pH (6.5) were achieved. These RNAs were the basis for LNP generation.

LNPs were prepared by mixing appropriate volumes of lipids in ethanol with a vRNA containing aqueous phase using a NanoAssemblr (Precision NanoSystems) microfluidic device followed by downstream processing. Using a flow ratio of 3:1 aqueous: organic phase, the solutions were combined using a microfluidic chip and 12 mL/min total flow rate. LNPs were dialyzed against a neutral pH buffer such as 1× PBS to remove ethanol and raise the pH. The resulting LNPs were concentrated using Amicon Ultra centrifugal filter units with 100,000 Da molecular weight cut-off (Millipore, Burlington, MA). RNA encapsulation was assayed using Quant-iT RiboGreen (Thermo Fischer Scientific, Waltham, MA) and a microplate reader (SpectraMax, San Jose, CA). Hydrodynamic size and PDI of the LNPs were analyzed by dynamic light scattering using a Zetasizer Nano ZS (Malvern Panalytical, Malvern, United Kingdom).

## vRNA transfection, infection and viral stock production
Viral stock production was done by vRNA transfection using Lipofectamine RNAiMax® (Thermo Fisher Scientific, Waltham, MA). Transfection reagent (1µg/ml) was added to NCI-H1299 cells seeded in a 12-well tissue culture plate at $1 \times 10^5$ cells/well. After 72 h post-transfection, supernatants were collected, centrifuged at $2000 \times g$ for 5 minutes, and filtered through a 0.45 µm filter. Filtered supernatant (100 µl) was used to infect a new 12-well plate seeded with NCI-H1299 cells. After 72 h, cells and supernatants were subjected to 3× freeze-thaw cycles, then centrifuged and filtered. Supernatant (1 mL) was then used to infect NCI-H1299 cells grown to 80% confluency in a two-chamber CellSTACK® (Corning, Corning, NY) tissue culture vessel in 250 ml of growth media. After 96 h post-infection, the cells and supernatants were 3× freeze-thawed, centrifuged, filtered, and concentrated using 100K Amicon® Ultra 15 Centrifugal Filter Units (Millipore, Burlington, MA) to a 10 ml final volume which was aliquoted and stored at −80 °C.

## Viral plaque titers and IC$_{50}$ protocols
To quantify SVV infectivity by IC$_{50}$ NCI-H446 cells were seeded into 96-well tissue culture plates at $1 \times 10^4$ cells/well and incubated at 37 °C, 5% CO$_2$. After 48 h, culture medium was aspirated and replaced with 10-

fold serial dilutions of infectious SVV in a 100 µL volume. After 48 h, cell viability of infected or mock-treated cells was measured with CellTiter-Glo® luminescent assay (Promega, Madison, WI) using a microplate reader (Molecular Devices, SpectraMax i3X minimax imaging cytometer, San Jose, CA). Raw data was converted to percentage survival relative to mock-infected. Values were graphed in GraphPad Software Prism 9.0 and analyzed using a non-linear sigmoidal plot with variable slope (asymmetric four-point linear regression) to generate IC$_{50}$ values. At least five technical repeats were analyzed for each sample to calculate IC$_{50}$.

To quantify SVV titer by plaque assay, $2 \times 10^5$ MCC14/2 cells/well were seeded into 12-well tissue culture plates. After 24 h, infectious virus was 10-fold serially diluted in serum-free media. Culture media from the cells was aspirated and replaced with 300 µL/well of diluted virus for an adsorption period of 1 h at 37 °C with gentle rocking. Virus samples were analyzed in technical triplicate. Post adsorption, 2 mL of pre-warmed 1% methylcellulose in media with 5% FBS was added to each well. After 48 h, wells were aspirated and stained with crystal violet solution. Discrete plaque-forming colonies were counted manually to determine titer.

To quantify CVA21 virus by plaque assay, $2 \times 10^5$ SK-MEL28 cells/well were seeded in a 24-well plate (or $2.5 \times 10^5$ NCI-H1299 cells/well in a 12 well plate). Mouse tissue homogenates were produced by resuspending pulverized frozen tissue samples in 2 µl PBS/mg of tumor. After mixing, samples were pelleted, and supernatants were recovered. Whole tissue homogenates were 10-fold serially diluted in serum-free media. After media was removed, 250 µl of the dilutions were added to each well. The plate was gently rocked at 37 °C for 1 h. Then, 1 ml of pre-warmed 1% methylcellulose in 5% FBS containing media was added as an overlay. Plates were incubated for 48 h before adding 250 µl crystal violet stain to each well; afterward, the overlay was removed and rocked at room temperature for 30 min. The plates were washed and allowed to dry to visualize plaques.

## SVV antisera and neutralization assay
Polyclonal rabbit SVV antisera were generated against UV-inactivated SVV at Maine Biotechnology Services (Portland, ME). To measure the neutralization titer, NCI-H446 cells were seeded into 96-well tissue culture plates at $1 \times 10^4$ cells/well in complete media and incubated at 37 °C, 5% CO$_2$. After 48 h, $1 \times 10^7$ TCID$_{50}$/mL of SVV was mixed with rabbit anti-SVV sera diluted in complete media. Serial two-fold dilutions of sera from 1:20 to 1:5120 were tested. Culture media was aspirated from the cells and replaced with 100 µl of diluted SVV per well. Cells were returned to incubation at 37 °C, 5% CO$_2$, and in vitro viability assays were performed at 48 h post-infection by adding 100 µL/well of CellTiter-Glo® 2.0 reagent (Promega, Madison, WI). Total luminescence (RLU) was measured on a plate reader (Molecular Devices, SpectraMax i3X minimax imaging cytometer, San Jose, CA). Raw data were converted to percentage survival relative to mock-infected, and values were graphed in GraphPad Software Prism 9.0.

## Flow cytometry analysis of hDLL3 expression
NCI-H82 cells were harvested by detaching with trypsin–EDTA (ThermoFisher, Waltham, Massachusetts), washed by centrifugation and resuspended in ice-cold phosphate buffer saline (PBS) containing 5% bovine serum albumin (BSA). One million cells per mL were incubated with 10 µg/mL of DLL3 specific phycoerythrin-conjugated antibody (Cat# 154003) or isotype control (Cat# 400607). Both antibodies were purchased from BioLegend (San Diego, USA), and data were acquired on a BD LSRFortessa using BD FACSDiva software and analyzed using FlowJo software.

## Mice, tumor models, and treatment
**Studies in xenografts and syngeneic tumor models.** In vivo experiments in xenograft tumor models NCI-H466, NCI-H82, NCI-H1299, and

SK-MEL-28 were conducted in 8–12-week-old NU/NU nude female mice (Charles River Laboratories, Wilmington, MA). N1E-115 murine neuroblastoma tumor model was established in 8–12-week-old A/J female mice (The Jackson Laboratory, Bar Harbor, ME). For studies in 4T1 and CT26 tumor models, 8–12-week-old female BALB/c mice (Charles River Laboratories) were used. Studies using the MC38 and CT-2A-luc tumor model were conducted in 8–12-week-old female C57BL/6 mice (Charles River Laboratories). Per IACUC regulations, mice with subcutaneous tumors were humanely euthanized once tumor volume reached 2000 mm³. In some cases, this limit has been exceeded the last day of measurement and the mice were immediately euthanized. Excepting these cases, no deviations from the approved protocol occurred and the maximal tumor volume was not exceeded. All animals had unlimited access to a sterile, pelleted rodent diet and reverse osmosis-purified water and were maintained on a 12:12 h light:dark cycle with access to environmental enrichment. All animal protocols for mice and non-human primates were approved by the Oncorus Institutional Animal Care and Use Committee (IACUC) and performed according to IACUC regulations.

To establish subcutaneous xenograft tumor models, $5 \times 10^6$–$1 \times 10^7$ viable tumor cells were injected into the right flank of nude mice in 100 μl of Matrigel (Corning, Glendale, AZ):PBS (Gibco, Gaithersburg) mixture (1:1 v/v). For the subcutaneous syngeneic N1E-115 tumor model, viable $5 \times 10^5$ N1E-115 cells were injected in 100 μl of Matrigel in PBS mixture (1:1 v/v) into the right flank of A/J mice. Treatment was initiated when tumors reached the pre-determined volume of $150 \pm 30$ mm³ for human xenograft models and $100 \pm 25$ mm³ for syngeneic tumor model. Animals were pair-matched based on tumor volume and randomly assigned to treatment arms. Synthetic-RNA viruses were dosed intravenously at doses ranging from 0.025 to 3.0 mg/kg at weekly intervals for a total number of up to 4 doses, as explained in figure legends.

Anti-mouse PD-1 antibody (clone RMP1-14, cat no. BE-0146, BioXCell, Lebanon, NH) or rat IgG2a isotype control (clone 2A3, cat no. BE0089, BioXCell, Lebanon, NH) were dosed intraperitoneally (IP) at a dose of 200 μg/mouse, administered 3 times every 3 days.

For the orthotopic SCLC tumor model, viable $5 \times 10^6$ NCI-H82 cells suspended in Matrigel:PBS mixture (1:1 v/v) were implanted into the left lung lobe via intra-thoracic injection. Treatment commenced 2 weeks post-orthotopic cell inoculation and consisted of 2 1.0 mg/kg IV doses of Synthetic-SVV-Neg or Synthetic-SVV administered 7 days apart. For survival analysis, mice were observed for pre-determined survival endpoints, which in the case of orthotopic lung tumors comprised of symptoms of lung disease and decreased body conditions (i.e., body weight loss exceeding 20%).

For the subcutaneous MC38 tumor model, viable $5 \times 10^5$ MC38 cells were injected in 100 μl of PBS into the right flank of C57BL/6 mice. Murine breast cancer (4T1) and colon carcinoma (CT26) models were established in BALB/c mice by subcutaneous injection of $1 \times 10^6$ cells in 100 μl of PBS into the right flank of the animals. When tumors reached 250 mm³, mice were humanely euthanized, tumors were collected, enzymatically dissociated and immunophenotyped, as described below. For orthotopic CT-2A-luc mouse glioma tumors were established in 11–12 weeks old female C57BL/6 mice by intracerebral injection of $5 \times 10^4$ CT-2A-luc cells suspended in 2 μl of PBS. Tumors were allowed to grow for 14 days; then, mice were humanely euthanized, tumors were collected, enzymatically dissociated and immunophenotyped, as described below.

**Efficacy of Synthetic-SVV in mice passively immunized to SVV.** Nude mice with subcutaneous NCI-H446 tumors were passively immunized to SVV by intraperitoneal injection of 1.5 mg of rabbit SVV anti-serum. Control animals received a corresponding amount of normal rabbit serum (Normal Rabbit Serum, ImmunoReagents, Inc., Raleigh, NC). Immunization cycle was repeated twice 7 days apart.

After 24 h post each adoptive serum transfer, control and passively immunized mice were intravenously treated with either $10^6$ PFU of SVV-virions or 0.1 mg/kg Synthetic-SVV. Per IACUC regulations, mice were humanely euthanized once tumor volume reached 2000 mm³. In some cases, this limit has been exceeded the last day of measurement and the mice were immediately euthanized. Excepting these cases, no deviations from the approved protocol occurred and the maximal tumor voume was not exceeded.

**Efficacy of Synthetic-SVV in SCLC PDX model.** To establish the SCLC PDX model, $2 \times 2$ mm fragments of LU5184 SCLC tumors (Crown Bioscience, San Diego) were implanted using a sterile trocar into 6–8-week-old female NOD SCID mice (The Jackson Laboratory, Bar Harbor, ME). Treatment commenced once tumors reached a pre-determined volume of approximately $150 \, \text{mm}^3 \pm 50 \, \text{mm}^3$. Animals were pair-matched based on tumor volume and randomly assigned to treatment groups. Synthetic-SVV and Synthetic-SVV-Neg were dosed in 2 intravenous 1.0 mg/kg doses administered weekly. Animals on treatment were observed daily for clinical manifestations of adverse events, body weight, and tumor volumes were recorded biweekly. As approved by Crown Bioscience's IACUC, mice were humanely euthanized once tumor burden reached 3000 mm³. No deviations from the approved protocol occurred and the maximal tumor volume was not exceeded.

For the pharmacodynamic analysis of virus replication in SCLC PDX tumors, tumor-bearing mice were treated with a single, intravenous dose of 1 mg/kg Synthetic-SVV or Synthetic-SVV-Neg. Five days post-treatment, tumors were collected and processed to evaluate negative-strand SVV RNA by RT-qPCR. Another group of mice was treated with 2 intratumor doses of $10^6$ PFU SVV-virions on Days 1 and 3. Two days post-second treatment, tumors were collected and processed to evaluate negative-strand SVV RNA by RT-qPCR.

**Efficacy of Synthetic-SVV in SCLC GEMM model.** SCLC GEMM model was established as described previously[33] at the Preclinical Modeling, Imaging & Testing Core (PMIT) at the Koch Institute. Briefly, 8 weeks of age RPM mice (Rb1 fl/fl Trp53 fl/fl fl/fl Myc LSL/LSL (RPM) mice containing a Cre recombinase regulatable MycT58A allele under control of a CAG promoter, The Jackson Laboratory Bar Harbor, ME) were anesthetized and infected with $10^6$–$10^8$ PFU of Ad5-Cgrp-Cre viruses (University of Iowa) by intratracheal instillation as described elsewhere[47]. According to Institutional Biosafety Committee guidelines, viruses were administered in a Biosafety Level 2+ room. Male and female mice were equally divided between treatment groups for all experiments. Development of SCLC orthotopic tumors was monitored by MicroCT. Tumors were collected, and a primary culture was established and maintained in DME medium (Gibco, Gaithersburg, MD) supplemented with 10% heat-inactivated FBS (Gibco, Gaithersburg, MD) and 1% penicillin/streptomycin (Gibco, Gaithersburg, MD). To establish subcutaneous tumor models, $5 \times 10^6$ viable tumor primary cells were injected into the right flank of nude mice in 100 μl of Matrigel (Corning, Glendale, AZ):DME (Gibco, Gaithersburg) mixture (1:1 v/v) into the right flank of RPM mice. Treatment was initiated when tumors reached the pre-determined volume of $100 \pm 30$ mm3. Synthetic-SVV and Synthetic-SVV-Neg were dosed in 2 intravenous 1.0 mg/kg doses administered weekly. Animals on treatment were observed daily for clinical manifestations of adverse events, body weight, and tumor volumes were recorded biweekly. Per IACUC's regulations, mice were humanely euthanized once tumor burden reached 2000 mm³. No deviations from the approved protocol occurred and the maximal tumor volume was not exceeded.

For the pharmacodynamic analysis of virus replication in SCLC GEMM tumors, tumor-bearing mice were treated with 2 IV doses of 1 mg/kg Synthetic-SVV or PBS. Five days post-treatment, tumors were collected and processed to evaluate negative and positive-strands SVV

RNA by RT-qPCR and transcriptional profiling utilizing NanoString Mouse PanCancer IO 360 Gene Expression Panel (Supplementary Data 1, the raw data on a per animal/sample basis and IO360 scores are included).

**Synthetic-SVV tolerability in A/J mice.** Tolerability was assessed in SVV-permissive A/J mice bearing subcutaneous N1E-115 tumors established as described hereinabove. Animals were IV dosed with 3 mg/kg of Synthetic-SVV and observed daily for clinical symptoms. Body weights were collected at least for 3 consecutive days after each treatment. Immune responses were analyzed at 6 and 24 h following IV treatment, whereas clinical pathology was evaluated at the end of the study. For biodistribution analysis tissues (spleen, liver, heart, lung, kidney, muscle, tumor) were collected after 30 min., 24 h and 7 days and analyzed by LC-MS. Viral replication was assessed in naïve A/J mice after a single IV treatment with 3 mg/kg of Synthetic-SVV. Tissues (spleen, liver, heart, lung, kidney) were collected after 24 h and 7 days and analyzed by plaque titer assay.

**Synthetic-CVA21 tolerability in hICAM1 transgenic mice.** Tolerability was assessed in CVA21-permissive huICAM1 transgenic mice after IV dose of Synthetic-CVA21 1.6 mg/kg. Animals were observed daily for clinical symptoms of adverse events and weighed for 5 consecutive days following treatment. Tissue pathology and CVA21 replication in treated animals were evaluated at 2- and 7-days post-treatment.

**Studies in non-human primates.** It is well established in the literature that the tolerability pharmacokinetics, the immune response, biodistribution, and tolerability differ greatly between mice and NHP with respect to lipid nanoparticles[48]. Tolerability in NHP was assessed using 3–7 years old naive male cynomolgus monkeys of Mauritius origin (Envigo, Indianapolis, IN). Animals were housed in a temperature and humidity-controlled environment (18–30 °C and 30–70%, respectively). Automatically controlled 12 h dark/light cycle was maintained, except for times when scheduled procedures were conducted. All animals had unlimited access to water, were fed NHP lab diet 250, and had access to environmental enrichment as outlined in standard operating procedures.

Synthetic-SVV at a dose of 1 mg/kg was administered in 60 min IV infusion via peripheral vein (cephalic or saphenous) in temporary restricted non-sedated animals. Treatment was repeated at biweekly intervals for a total number of 3 doses. The first day of dosing was designated as day 1. For PK analysis, blood samples were collected at the end of infusion (EOI), 5 min, 0.5, 2, 8, 12, 24, and 48 h. Ionizable lipid (OC) plasma concentration was determined via LC-MS. Non-compartmental pharmacokinetic evaluation was conducted using Phoenix WinNonlin software version 8.3 (Certara, Princeton, NJ). For clinical pathology analysis, blood samples were collected pre-dose, as well as 2, 6, and 24 h post each treatment. Hematology, clinical chemistry, cytokine, and complement C5b9 component plasma levels were evaluated. Tissue samples (cerebrum, midbrain, cerebellum, medulla/pons, lumber spinal cord, heart, liver, kidney, lung, spleen, mesenteric lymph node, skeletal muscle) for biodistribution analysis were collected 24 h following 3rd IV treatment, flash frozen, and evaluated for the presence of positive-strand SVV RNA via RT-qPCR analysis.

**Clinical chemistry and hematology.** Clinical chemistry was analyzed using either Randox Imola analyzer (Randox, Kearneysville, WV) or hematology analysis conducted in Siemens Advia 120 automated analyzer (Siemens Healthcare GmgH, Erlangen, Germany).

**Evaluation of cytokine and complement C5b9 levels in plasma.** Analysis of pro-inflammatory cytokines plasma levels was conducted using NHP and mouse-specific multiplex assay (K15056D and K15048D, respectively; Meso Scale Diagnostics, Rockville, MD). In addition, cynomolgus monkey MCP-1 and TNFα were analyzed using single analyte assay kits (K156UCK, K156NND, respectively; Meso Scale Diagnostics). Similarly, mouse MCP-1 was analyzed using single-plex assay kit (K152NN, Meso Scale Diagnostics).

Mouse C3 complement component levels were analyzed using Mouse Complement C3 ELISA Kit (ab157711, Abcam, Waltham, MA). C5b9 levels in cynomolgus monkey plasma were analyzed using Human C5b9 ELISA Kit (558315, BD, Franklin Lakes, NJ).

### SVV and CVA21 RNA negative-strand specific RT-qPCR

**Tumor and tissue RNA extraction.** Samples were kept frozen during the entire procedure preceding RNA extraction using dry ice and liquid nitrogen to flash freeze. Samples were pulverized using Cp02 cryo-PREP Automated Dry Pulverizer (Covaris, 500001, Covaris, Woburn, MA) for SVV- and CVA21-treated tumor samples. For SVV samples, 10 mg of pulverized sample was weighed and transferred to a 2 mL microcentrifuge tube (Sample Tube RB, QIAGEN 990381, Hilden, Germany). Buffer RLT Plus with B-mercaptoethanol (600 μL) was added to each sample and lysed. The remaining steps were performed using the QIAcube (QIAGEN, Hilden, Germany), following the manufacturer's protocol for QIAGEN RNeasy Plus Mini Kit (QIAGEN 74134) under the section for Purification of Total RNA from Animal Tissues. RNA samples were treated with DNAseI (RNAse-free) (New England Biolabs, #M0303S, Ipswich, MA) after extraction.

For CVA21 samples, 10 mg of pulverized sample was weighed and transferred to a 1.5 mL microcentrifuge tube. Lysis Buffer/Proteinase K mixture (400 μL) (RNAdvance Tissue Kit, Beckman Coulter, A32649, Pasadena, CA) was added to each sample. Samples were incubated at 37 °C for at least 30 min to lyse samples completely. The remaining steps were performed using the Biomek i5 (Beckman Coulter, B87583, Pasadena, CA) following manufacturer's protocol (RNAdvance Tissue Kit). The RNAdvance Tissue Kit includes a DNase I treatment step.

**cDNA synthesis and quantitative PCR analysis.** RNA samples were normalized to an equal input for cDNA synthesis. Virus negative-strand specific primers were used to synthesize cDNA. The SVV negative-strand specific primer 5'-GCGCAAATTCGTCCAAAACAACGAC-3', SVV positive-strand specific primer 5' ACATAGAAACAGATTGCAGCTTCTC -3', and the CVA21 negative-strand specific primer 5'-AGACTACGG ACTGACCATGACTC TTAGGACGCTTTTACTGAGAAC -3' were synthesized by Integrated DNA Technologies (IDT, Coralville, Iowa). For both SVV and CVA21, cDNA synthesis was performed using SuperScript IV First-Strand Synthesis System (Invitrogen, 18091200, Carlsbad, CA) and their respective specific primers.

Quantitative PCR (qPCR) analysis was performed using TaqMan probe chemistry. qPCR reactions (20 μL) were made containing 10 μL of TaqMan Fast Advanced Master Mix (Applied Biosystems, 4444557, Foster City, CA), 1 μL of TaqMan Gene Expression Assay, FAM probe (Applied Biosystems, 4332078), 4 μL of nuclease-free water, and 5 μL of diluted cDNA. TaqMan Gene Expression Assay probes were made customized for either SVV or CVA21. SVV probe 5'-TGGAAGCCA TGCTCTCCTACTTCA-3', forward primer 5'-CGACGGCTTATACAAA CCAGTTA-3', reverse primer 5'-AGCTTCTCGAGTAGTGTTCCT-3' and CVA21 probe 5'-TGCCTATGGTGATGACGTGATAGCT-3', forward primer 5'- GAGAACCTACAAGGGCATAGAC-3', and reverse primer 5'-TAGGAGACTAGCGTCAACCT-3' were custom ordered through Applied Biosystems (ThermoFisher Scientific, Waltham, MA). qPCR parameters 95 °C for 20 s followed by 40 cycles at 95 °C for 1 s and 60 °C for 30 s. Each qPCR assay contained technical triplicates for each standard and sample.

**SVV fluorescent in situ hybridization.** Tumors were harvested at the indicated timepoints, bisected, fixed in 10% buffered formalin for 24 h

at room temperature, and paraffin-embedded. RNAscope FISH of SVV negative and positive RNA strands was performed at Advanced Cell Diagnostics (ACD, Newark, CA). Standard RNAscope LS Multiplex fluorescent pretreatment conditions were used. Briefly, epitope retrieval was performed at 15 min at 95 °C, followed by protease III treatment for 15 min at 40 °C. Samples were first evaluated for quality using the positive and negative reference controls for specificity and sensitivity (ACD Catalog # 313908 and 320758, respectively). Custom SVV probes were designed for both the positive and negative strands (ACD Catalog # 819848 and 819858-C2, respectively) and were confirmed to be specific. All samples passed QC with positive control staining, and little to no background staining was observed. RAW data was analyzed with 3DHISTECH CaseViewer Software (3DHISTECH Ltd, Budapest, Hungary).

**Quantification of OC-lipid by LC-MS.** OC lipid concentration was determined via LC-MS analysis. Standard stock solution at 100 ng/ml was prepared in 1% formic acid in methanol. For plasma sample analysis, standard stock was diluted in plasma to 200, 180, 120, 40, 20, 5, 1, and 0.5 µg/ml, whereas for tissue analysis a standard curve was generated at 80, 72, 48, 16, 8, 2, 0.4, and 0.2 mg/ml in tissue homogenate. Five µl of plasma sample were mixed with 120 µl of internal standard control (IS-C), vortexed for 3 min and centrifuged at 12,500 RPM at 5 °C for 4 min. Sixty µl of resulting supernatant was mixed with 140 µl of water:acetonitrile solution (3:7 v/v) and shaken for 5 min. An aliquot of 0.6 µl of resulting supernatant was injected for LC-MS analysis.

Twenty-five µl of tissue homogenates obtained from tissue samples homogenized in water at 1:1 mμm ratio were mixed with IS-C, vortexed for 3 min, then mixed with 140 µl of water and acetonitrile solution (3:7 v/v), and shaken for 5 min. An aliquot of 0.6 µl of resulting supernatant was injected for LC-MS analysis.

A Waters Zevo TQs system equipped with a MRM detector was used for the chromatographic separation. Separation was carried out on a XB-C8 column (50 mm × 2.1 mm, 3.6 µm) with an injection volume of 0.6 µl. The mobile phases consisted of 0.1% formic acid in water (A) and 0.1% formic acid in acetonitrile: isopropyl alcohol (66:34 v/v) (B). Samples were analyzed using flow rate of 0.5 ml/min, and the following gradient program: 54–100% B (0–1.51 min), 100% B (1.51–1.8 min), 100–54% B (1.80–1.81 min), and 54% B (1.81–2.8 min). Ionization was achieved using electrospray ionization (ESI) in positive mode with MRM detection.

**Immunohistochemical analysis of orthotopic tumor burden.** Lungs were collected from treated animals 10 days post-second Synthetic-SVV dose and were fixed in 10% buffered formalin for 24 h, paraffin processed, and sectioned at the level of mainstem bronchi. For IHC detection of DLL3, sections were stained with rabbit antibody specific to human DLL3 (0.5 µg/ml, Clone E3J5R, Cell Signaling Technology, Danvers, MA), digitally scanned, and subjected to morphometric analysis using QuPath software.

**Transcriptional (NanoString) analysis.** Flash-frozen tumors ($n$ = 5 mice per treatment group) were pulverized, and RNA was extracted using QIAcube (QIAGEN, Hilden, Germany), following the manufacturer's protocol for QIAGEN RNeasy Plus Mini Kit (QIAGEN 74134) under the section for Purification of Total RNA from Animal Tissues. Total isolated RNA (60 ng) was mixed in a final volume of 25 µl with 3' biotinylated capture probe and 5'-reporter probes from Mouse Pan-Cancer IO 360 Gene Expression Panel. Hybridization was conducted at 65 °C for 16 h. Hybridized samples were isolated on the NanoString nCounter preparation station where the excess probe was removed, and hybridized complexes of probes and target RNA sequences were immobilized on the cartridge. The cartridge bound samples were then analyzed using nCounter digital analyzer. Data collection was

carried out on the nCounter Digital Analyzer (NanoString Technologies) following the manufacturer's instructions to count individual fluorescent barcodes and quantify target RNA molecules present in each sample. For each assay, a high-density scan (600 fields of view) was performed. Nanostring nCounter™ gene expression analysis relative mRNA copy number on 770 cancer and immune system-related genes was quantified on the NanoString nCounter™ system (NanoString Technologies), according to the manufacturer's instructions. Results were analyzed using the nSolver Analysis Software 4.0 (NanoString). Upon background subtraction (removal of the geometric mean of negative controls), each sample was normalized first to the geometric mean of positive controls and then to the geometric mean of reference genes.

**Immune profiling of tumors.** For flow cytometry, tumors were collected on Day 14 after the first dose. Tumors were weighed and then cut into small pieces before being disaggregated into single-cell suspensions using a Miltenyi GentleMACs Octo-dissociator with heaters according to the manufacturer's instructions. A T/NK cell and a myeloid cell flow panel were performed using the following reagents and Ab clones: A LIVE/DEAD™ Fixable Red Dead Cell Stain Kit for 488 nm excitation (Invitrogen, Catalog # L32102) was used to assess the viability. Surface cell staining was performed by using BD Stain Buffer (Becton-Dickinson Catalog # 554656) with the antibodies for mouse CD45 (30-F11, Catalog # 103154), CD3ε (17A2, Catalog # 100218), CD8a (53-6.7, Catalog # 100730), CD4 (RM4-5, Catalog # 100552), CD25 (PC61, Catalog # 102041) CD69 (REA937, Catalog # 130-115-461), CTLA-4 (UC10-4B9, Catalog # 106323), NKP46 (29A1.4, Catalog # 137612), KLRG1 (2F1, Catalog # 138409), CD127 (A7R34, Catalog # 135035), PD-1 (29F.1A12, Catalog # 135216), MHCII (M5/114.15.2, Catalog # 107635), Ly6C (H,1.4, Catalog # 128037), CD206 (C068C2, Catalog # 141720), CD86 (GL-1, Catalog # 105037) and PD-L1 (10F.9G2, Catalog # 124312). Foxp3/Transcription Factor Staining Buffer Set (eBioscience Catalog # 00-5523-00) was used for FOXP3 (150D, Catalog # 320008) and CTLA-4 (UC10-4B9, Catalog # 106323) intracellular staining. All antibodies were purchased from BioLegend (San Diego, USA), except CD69 Miltenyi Biotech (Bergisch Gladbach, Germany). Antibody dilutions were as follows: CD45-APC-Fire750 (1:400), CD3-PerCP-Cy5.5 (1:100), NKp46-BV421 (1:200), CD4-BV785 (1:800), CD8-AF700 (1:400), CD25-BV711 (1:200), Foxp3-PE (1:200), CTLA4-BV605 (1:200), PD1-PE-Cy7 (1:200), CD127-BV510 (1:100), KLRG1-FITC (1:400), CD11b-BV650 (1:400), MHCII-BV510 (1:200), Ly6C-BV711 (1:200), CD206, PE-Cy7 (1:100), CD86-BV605 (1:100) and PDL1-APC (1:100). Data were acquired on a BD LSRFortessa using BD FACSDiva software and analyzed using FlowJo software. For the T cell and NK panel, cells were first gated for time (SSC-A vs. Time), lymphocytes (SSC-A vs. FSC-A), and singlets (FSC-H vs. FSC-A). The lymphocyte gate was further analyzed for their uptake of the Live/Dead stain to determine live vs. dead cells and CD45 expression. Then, the cells were gated on CD3 vs. NKP46 to select T cells or NK cells. For the T cells, the population was gated for CD4 vs. CD8, and the CD4 T cells were further gated for CD25+ and FOXP3+ to analyze the Treg population. NK cells were gated for NKP46+ and CD3−. For the myeloid panel, cells were first gated for time (SSC-A vs. Time) then lymphocytes (SSC-A vs. FSC-A), and singlets (FSC-H vs. FSCA). The lymphocyte gate was further analyzed for their uptake of the Live/Dead stain to determine live vs. dead cells and CD45 expression. Then, macrophages were gated using SSC-A vs. CD11b+ and subsequently on MHCII+Ly6C-subset. To distinguish between M1 and M2 macrophages, we used CD206-CD86+ for M1 and CD206+CD86- for M2. Enumeration of cells per mg of tumor was calculated using 123Count eBeads (Invitrogen, Catalog # 01-1234-42, Carlsbad, CA). A fixed volume of beads with a known concentration was added to each well prior to analysis by flow cytometry. The number of beads

in the gated fraction was then used to calculate cell number using the following equation:

$$\text{Absolute count}\left(\frac{\text{cells}}{\mu L}\right) = \frac{(\text{cell count } x \text{ eBead volume})}{(\text{eBead count } x \text{ cell volume})} \times \text{eBead concentration}$$

### Reporting summary

Further information on research design is available in the Nature Research Reporting Summary linked to this article.

## Data availability

The data that support the findings in this study are available within the Article, Supplementary Information or Source Data file. Source data are provided with this paper.

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

## Acknowledgements

This study was sponsored by Oncorus, Inc. The authors acknowledge and thank Sofie Denies for statistical analysis, James B. Rottman (Athenaeum Pathology Consulting) for morphometric analysis. The authors also thank Chastity Bradley, PhD of BioMed Writers, LLC, for her editorial review and assistance with manuscript submission.

## Author contributions

Conceptualization: E.M.K., M.F., C.Q., and L.L. Methodology: E.M.K., A.D., J.H., L.K., A.D.A., J.D.B., J.S.L., J.J., S.F., M.H., E.L.M., D.W., T.F., A.A.R., L.H., D.D., J.S., S.A., J.D., M.S., B.B.H., C.Q., and L.L. Software: E.M.K., A.D., J.S.L., S.F., and D.W. Validation: E.M.K., A.D., J.H., L.K., A.D.A., J.D.B., J.S.L., J.J., S.F., M.H., E.L.M., D.W. T.F., A.A.R., L.H., D.D., J.S., S.A., J.D., and M.S. Formal analysis: E.M.K., A.D., J.D.B., J.S.L., J.J., S.F., B.B.H., C.Q., L.L. Writing – review & editing: E.M.K., A.D., A.D.A., J.D.B., J.S.L., S.F., J.D., M.S., B.B.H., C.Q., and L.L. Supervision: E.M.K., A.D., S.F., B.B.H., C.Q., and L.L. Project administration: E.M.K., C.Q., and L.L. Approval of final draft: E.M.K., A.D., J.H., L.K., A.D.A., J.D.B., J.S.L., J.J., S.F., M.H., E.L.M., D.W. T.F., A.A.R., L.F., D.D., J.S., S.A., J.D., M.S., B.B.H., T.T.A., M.F., C.Q., and L.L.

## Competing interests

E.M.K., A.D., J.H., L.K., A.D.A., J.D.B., and J.S.L. (at the time the study was conducted), J.J., S.F. (at the time the study was conducted), M.H., E.L.M., D.W., T.F., A.A.R. (at the time the study was conducted), L.H. (at the time the study was conducted), D.D. (at the time the study was conducted), J.S. (at the time the study was conducted), S.A. (at the time the study was conducted), J.D., M.S. (at the time the study was conducted), B.B.H. (at the time the study was conducted), M.F. (at the time the study was conducted), T.T.A., C.Q., L.L. (at the time the study was conducted) are all employees of Oncorus, Inc. No other authors have any competing interests to disclose.
