## [Peer Review File · Nature Communications]

Development of intravenously administered synthetic RNA virus immunotherapy for the treatment of cancerEDITORIAL NOTE: This manuscript has been previously reviewed at another journal that is not operating a transparent peer review scheme. This document only contains reviewer comments and rebuttal letters for versions considered at Nature Communications.

REVIEWER COMMENTS

Reviewer #2 (Remarks to the Author):

The authors have not satisfactorily addressed my previous major concerns. I remain of the opinion that the SVV-001 virus used in Fig. 3 is a wrong control and that the proper control would instead have been the SVV-S177A virus (the same one that is encapsulated in the nanoparticles). Furthermore, this virus should have been used at the same concentrations in terms of viral genome as the synthetic virus and not 10 times less concentrated like the authors have done. Although Fig. S8 shows that passive immunization neutralizes both SVV-001 and SVV-001-IRES-S177A in a similar manner, the authors cannot deny that the SVV-001 and the SVV-S177A are still different viruses, SVV-S177A being significantly more potent than SVV-001 (e.g. authors have shown in Fig. S3 C that the synthetic SVV-S177A blocks tumour growth more efficiently than SVV-001 and ref 21 reports that the S177A modification significantly increases SVV proliferation). SVV-S177A is more infectious than SVV-001 and thus can replicate faster and kill more efficiently than SVV-001 tumour cells before neutralization by the immune system takes place. Based on the results shown in Fig. 3, I also disagree with authors' conclusion (author's response letter) that in the absence of neutralizing antibodies the two viruses are equally potent. The results show that in the absence of neutralizing antibodies, tumours did not develop in both SVV virus and Synthetic-SVV treated groups. This does not necessarily mean that the two viruses are equally potent. It is still possible that one virus is superior to the other but it is impossible to appreciate this superiority because under the chosen experimental conditions, the weaker treatment is already sufficient to produce the complete tumour growth abrogation (e.g. if the Synthetic virus would have been ten time more potent, similar results would have been generated). So the "key finding that the potential benefit of the Synthetic platform is that the efficacy is maintained in the presence of neutralizing serum to the virus" in light of these results still remains to be consolidated.

Reviewer #4 (Remarks to the Author):

Comment NCOMMS-22-09689-T

Thank you for the opportunity to review this interesting manuscript. As requested, I have focused on the non-human primate aspects of the study and on the response of the authors to the questions raised by referee #1.

The question raised by the reviewer is a valid one. I would be satisfied by the answer of the authors. Indeed, in Suppl Tab 1, we see a 2,5-3x higher exposure of the synthetic-SVV as compared to mice which could justify the 3x higher administration dose.

It is known that many aspects like pharmacokinetics, distribution and especially immunological responses differ significantly between mice and non-human primates (as closest relative to human). Therefore, I would suggest that the second statement of the authors: "It is also well established in the

literature that", would be added in the Methods section somewhere around lines 524-525, with one or two key literature references showing the differences in tolerability of lipid nanoparticles in NHP vs Mice. This extra information will give the justification of the differences in concentrations of synthetic-SVV administrations in both animal species.

Point-by-Point Response to Reviewers

Reviewer #2:

I remain of the opinion that the SVV-001 virus used in Fig. 3 is a wrong control and that the proper control would instead have been the SVV-S177A virus (the same one that is encapsulated in the nanoparticles). Furthermore, this virus should have been used at the same concentrations in terms of viral genome as the synthetic virus and not 10 times less concentrated like the authors have done.

Although Fig. S8 shows that passive immunization neutralizes both SVV-001 and SVV-001-IRES-S177A in a similar manner, the authors cannot deny that the SVV-001 and the SVV-S177A are still different viruses, SVV-S177A being significantly more potent than SVV-001 (e.g. authors have shown in Fig. S3 C that the synthetic SVV-S177A blocks tumour growth more efficiently than SVV-001 and ref 21 reports that the S177A modification significantly increases SVV proliferation). SVV-S177A is more infectious than SVV-001 and thus can replicate faster and kill more efficiently than SVV-001 tumour cells before neutralization by the immune system takes place.

As requested, we have rescued the virus encoded by Synthetic SVV (SVV(S177A-IRES)) and repeated the experiment with this virus at a dose 10-fold lower than the experiment previously included, and we have added this to body of the manuscript. We have moved the previous figure including the SVV-001 virus control and the 1 mg / kg dose so supplemental figure 8. As previously shown, we observe that with both viruses, SVV-001 and the virus encoded by Synthetic SVV, SVV(S177A-IRES2), that efficacy is completely abrogated the SVV antisera shown to neutralize both viruses, and that in both cases tumor growth for these conditions are identical to PBS. We also observe comparable highly statistically significant efficacy in the same model for Synthetic SVV regardless of the sera (neutralizing or control) employed, for 2 complete animal experiments at 2 doses.

Based on the results shown in Fig. 3, I also disagree with authors' conclusion (author's response letter) that in the absence of neutralizing antibodies the two viruses are equally potent. The results show that in the absence of neutralizing antibodies, tumours did not develop in both SVV virus and Synthetic-SVV treated groups. This does not necessarily mean that the two viruses are equally potent. It is still possible that one virus is superior to the other but it is impossible to appreciate this superiority because under the chosen experimental conditions, the weaker treatment is already sufficient to produce the complete tumour growth abrogation (e.g. if the Synthetic virus would have been ten time more potent, similar results would have been generated).

So the "key finding that the potential benefit of the Synthetic platform is that the efficacy is maintained in the presence of neutralizing serum to the virus" in light of these results still remains to be consolidated.

It is important to clarify that direct comparison of the potency of either SVV virus employed to Synthetic SVV is not the objective of this experiment as they are dramatically distinct modalities, one is a picornavirus, and the other is LNP containing the positive sense vRNA. They differ with respect to stability in plasma, delivery of the RNA to the cytoplasm, and cellular entry tropism; however, both are therapeutically effective at a well-tolerated dose in the absence of the neutralizing antibodies. The objective of this experiment was to test this hypothesis that LNP encapsulated vRNA could initiate replication in a tumor in the presence of neutralizing antisera, therefore supporting the therapeutic premise that this agent could be dosed effectively and repeatedly as we have shown. We have done this with the appropriate controls with two separate experiments at two therapeutically relevant doses. We appreciate the commentary of the reviewer and feel that this discussion (we have added commentary in the discussion, see lines 251-254) and the addition of the requested data greatly strengthens the manuscript.

Reviewer #4

Thank you for the opportunity to review this interesting manuscript. As requested, I have focused on the non-human primate aspects of the study and on the response of the authors to the questions raised by referee #1.

The question raised by the reviewer is a valid one. I would be satisfied by the answer of the authors. Indeed, in Suppl Tab 1, we see a 2,5-3x higher exposure of the synthetic-SVV as compared to mice which could justify the 3x higher administration dose.

It is known that many aspects like pharmacokinetics, distribution and especially immunological responses differ significantly between mice and non-human primates (as closest relative to human). Therefore, I would suggest that the second statement of the authors: "It is also well established in the literature that ...", would be added in the Methods section somewhere around lines 524-525, with one or two key literature references showing the differences in tolerability of lipid nanoparticles in NHP vs Mice. This extra information will give the justification of the differences in concentrations of synthetic-SVV administrations in both animal species.

We have added this commentary and an appropriate reference as requested. We agree and thank the reviewer for the helpful suggestion.

REVIEWERS' COMMENTS

Reviewer #2 (Remarks to the Author):

The authors have adequately addressed all my remaining concerns.